# Accounting for AI and Users Shaping One Another: The Role of Mathematical Models

**Sarah Dean**                                           *sdean@cornell.edu*
*Department of Computer Science*
*Cornell University*

**Evan Dong**                                            *edong@cs.cornell.edu*
*Department of Computer Science*
*Cornell University*

**Meena Jagadeesan**                                     *mjagadeesan@berkeley.edu*
*Department of EECS*
*University of California, Berkeley*

**Liu Leqi**                                             *leqiliu@utexas.edu*
*Department of IROM*
*University of Texas, Austin*

**Reviewed on OpenReview:** *https://openreview.net/forum?id=UkP4DhrJt1*

## Abstract

As AI systems enter into a growing number of societal domains, these systems increasingly shape and are shaped by user preferences, opinions, and behaviors. However, the design of AI systems only sometimes accounts for how AI and users shape one another. In this survey paper, we discuss the development of *formal interaction models* which mathematically specify how AI and users shape one another. Formal interaction models can be leveraged to (1) specify interactions for implementation, (2) monitor interactions through empirical analysis, (3) anticipate societal impacts via counterfactual analysis, and (4) control societal impacts via interventions. The design space of formal interaction models is vast, and model design requires careful consideration of factors such as style, granularity, mathematical complexity, and measurability. Using content recommender systems as a case study, we critically examine the nascent literature of formal interaction models with respect to these use-cases and design axes. More broadly, we call for the community to leverage formal interaction models when designing, evaluating, or auditing any AI system which interacts with users.[1]

## 1 Introduction

Machine learning enables the creation of platforms and products which are data-driven, adaptive, and intelligent. When these AI systems[2] are deployed into the world, it is common for these systems to shape their users, and in turn for users to shape these systems. For example, consider *content recommender systems*, which provide algorithmic curation and personalization. Content recommender systems interact with both viewers and creators; these interactions can shape viewer preferences (Leqi et al., 2023) and shape the landscape of content available on the platform (Meyerson, 2012), which in turn affects downstream recommendations. Another example is *user-facing decision-support systems*, which have been integrated into hiring pipelines (Commission et al., 2022), credit score assessment (Sadok et al., 2022), and resource allocation

---

[1]Authors in alphabetical order.

[2]An AI system refers to any algorithmic or learning system which transforms data into predictions, decisions, or other output forms (e.g., through optimizing a loss function).

(Obermeyer et al., 2019); these systems have shaped the actions and outcomes of users (Björkegren et al., 2020). More recently, *large language models (LLMs)* have started to interact with ecosystems of consumers and service-providers, which has created new avenues for shaping the world at large (Pan et al., 2024).

The fact that AI systems and users shape one another can lead to unanticipated ramifications for platforms, users, and society at large. In established domains, these consequences have been well documented. For example, content recommender systems can lead to clickbait at the level of the platform (Meyerson, 2012), addiction at the level of individual users (Hasan et al., 2018), and growing radicalization at the level of society (Haroon et al., 2022). As another example, user-facing decision-support systems (e.g., in hiring pipelines, credit score assessment, and resource allocation) can lead to strategic behavior (Rosenfeld, 2024) and can affect access to opportunities and fairness objectives (Reader et al., 2022; Pagan et al., 2023). In fact, even in nascent domains, AI systems have already started to exhibit such dynamics. For example, LLM-based chatbots can produce emotionally charged and aggressive language (Perrigo, 2023), and text generation models enable the mass production of SEO spam (Notopoulos, 2024). Furthermore, the prospect of autonomous cars brings these concerns from the digital to the physical world, where smart routing algorithms may disrupt driving norms or traffic flow (Sadigh et al., 2016), and incentivize antagonistic behaviors by humans (Paul, 2023).

The core conceptual problem is that AI system design and evaluation does not appropriately account for how these systems and users shape one another. In fact, this conceptual problem surfaces in many learning paradigms. For example, in supervised learning settings, the traditional learning paradigm of training on a large static dataset fails to account for how the AI system changes the environment in which it operates. At deployment, the AI system can trigger a unforeseen distribution shift, which can lead to unintended impacts on performance and on society at large. To some extent, approaches based on reinforcement learning (RL) attempt to account for such shifts. However, the state representations, reward function, and transition models—which are implemented in simulators or learned from data in model-based RL—often fail to capture key aspects of interactions and dynamics between the AI system and users. As a result, at deployment, the RL policy can also shape the environment in unforeseen ways and similarly trigger unintended impacts (Gilbert et al., 2022; 2023).[3]

Towards addressing the shortcomings of AI system design and evaluation, we highlight an emerging strategy based on the development of *formal interaction models*, i.e., mathematical models which formalize how AI and users shape one another. We define a formal interaction model as a coupled dynamical system between the AI system and users (Section 2.1, Figure 1b). Formal interaction models can be leveraged to improve AI system design and evaluation, as illustrated by the following use-cases: (1) specifying interactions for implementation, (2) monitoring interactions through empirical analysis, (3) anticipating societal impacts via counterfactual analysis, and (4) controlling societal impacts via interventions (Section 2.2). The design space of formal interaction models is vast, and there is no "one-size-fits-all" approach for designing a model: which approach is most appropriate is use-case dependent. Model design thus requires careful consideration of design axes such as style, granularity, mathematical complexity, and measurability (Section 2.3).

Taking content recommendation as a case study, we critically examine the nascent literature of formal interaction models with respect to these use-cases and design axes (Section 3). The formal interaction models in this literature capture how recommender systems can shape users—from viewer preferences to content creator strategies—but different works formalize these models in varying languages (e.g., nonlinear dynamics, game theory, bandits, and behavioral psychology). To compare the formal interaction models proposed in different works, we cast each formal interaction model within the coupled dynamical systems language (Figure 2). We identify that the literature primarily focuses on anticipating and controlling societal impacts, suggesting that there is room for future work focused on specifying and monitoring interactions. We further observe that the models of each interaction type are fairly homogeneous in style, and that models of viewer interactions and creator interactions have been largely separate; this motivates future work which broadens the scope of these models.

---

[3]See Section 2.1 for additional discussion of how standard learning paradigms fail to appropriately account for how AI and users shape one another. The discussion is specialized to content recommender systems for concreteness.

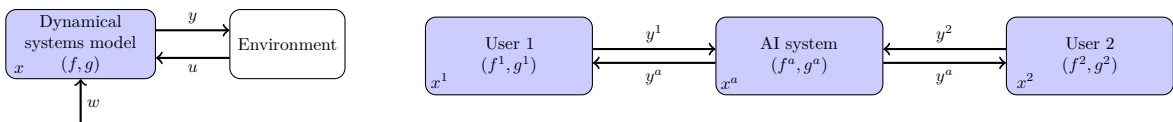

(a) Model and environment interaction      (b) Formal interaction model as a coupled dynamical system

Figure 1: *Left:* a dynamical system where a model interacts with an unspecified environment. *Right:* a formal interaction model as a coupled dynamical system between an AI system and users. The variable $x$ captures the internal state, the function $f$ captures the state update, the function $g$ captures the measurement equation, the variable $u$ denotes inputs, the variable $y$ denotes outputs (which also serve as inputs, in the case of the coupled dynamical system), and the variable $w$ captures noise or other disturbance (Section 2.1).

More broadly, we advocate for formal interaction models as a principled approach to account for AI systems and users shaping one another across a wide range of domains (e.g., content recommender systems, LLM-based ecosystems, decision-support systems, autonomous vehicles, etc). We are inspired by the success of formal interaction models in *mechanism design* and *physical systems* (Appendix A), and we expect that formal interaction models—if appropriately leveraged—can similarly contribute to AI system design. We call for the community to leverage formal interaction models when building, evaluating, deploying, and auditing AI systems: only when we properly account for how users and AI systems shape one another will we be able to build reliable AI systems with positive societal ramifications.

## 2 Formal Interaction Models

We investigate the role of formal (mathematical) models in accounting for interactions between AI systems and users. We focus a class of models—which we call *formal interaction models*—which specify how an AI system *shapes* the internal state of individual users and how these users *shape* the AI system. We provide an overview of formal interaction models (Section 2.1), present use cases of formal interaction models (Section 2.2), and discuss how model design requires careful consideration along several axes (Section 2.3).

### 2.1 Overview of Formal Interaction Models

We first justify the need for formal interaction models, using content recommender systems as a motivating example. Then, we define formal interaction models both conceptually and as a coupled dynamical system. Finally, we give examples of formal interaction models for user-AI interactions across many domains.

**Motivating Example: Content Recommendation.** While content recommendation was initially designed to help individuals navigate a "deluge" of articles, songs, and videos sent over email lists (Konstan et al., 1997; Hill et al., 1995; Shardanand & Maes, 1995), modern content recommendation is performed by large online platforms. On today's online platforms—such as YouTube, Netflix, and TikTok—viewer and creator populations are within closed gardens and can be shaped by the algorithms deployed by the platform (Gillespie, 2014; Seaver, 2022). In particular, the specifics of a content recommendation model can shape the *internal state of viewers* such as viewers preferences, behaviors, or participation decisions; the content recommendation model can also shape the *internal state of creators* such as creator decisions about what content to create, which affects the content landscape. Moreover, these changes to viewer preferences and the content landscape alter the characteristics of the online platform, contributing to societal level impacts. (See Section 3 for further discussion of how the recommender system can shape viewers and creators, and vice versa.)

However, standard approaches for designing content recommender systems fail to appropriately account for how AI and users shape one another. For example, consider *two-tower embedding models*—based on matrix factorization (e.g. Hu et al., 2008) or deep learning variants (e.g. Yi et al., 2019; Yang et al., 2020)—where the recommender systems learns embeddings for content and embeddings for viewer preferences, and assigns recommendations based on these embeddings. The main issue is that these architectures implicitly treat the viewer preference embeddings and content embeddings as fixed, and thus fail to account for how the

recommender system shapes viewer preferences and the creator incentives. As another example, consider *model-free reinforcement learning (RL) approaches*. One issue is that RL approaches make extensive use of simulators (e.g. Ie et al., 2019) which often make simplifying assumptions about how recommender systems shape viewers and creators. In fact, most simulators entirely ignore how the recommender system shapes creator decisions, and treat the landscape of content available on the platform as fixed. Even when simulators do account for how the recommender system shapes viewers, RL approaches optimize aggressively for reward signals and can unintentionally manipulate viewer preferences in the process (e.g. Carroll et al., 2022).

As a result of these conceptual shortcomings, deployed recommender systems have induced unintended societal ramifications for the platform (e.g., performance degradation due to clickbait (Meyerson, 2012)), users (e.g., addiction (Hasan et al., 2018)), and even society at large (e.g., radicalization (Haroon et al., 2022)). This highlights the need for better accounting of how the recommender system and users shape one another (Boutilier et al., 2023; Leqi & Dean, 2022). As we describe in more detail below, formal interaction models are designed to capture these dynamics.

**High-Level Definition of a Formal Interaction Model.** *Formal interaction models* are mathematical models which capture how an AI system shapes users and how users shape the AI system. An AI system refers to any algorithmic or learning system which transforms data into predictions, decisions or other output forms (e.g., through optimizing a loss function). A user refers to any human who interacts with the AI system (directly or indirectly). Formal interaction models can draw principles from *many disciplines* (e.g., dynamical systems, online learning, game theory, industrial organization, behavioral psychology, causal modelling, etc.), but nonetheless share a common high-level structure that we describe below.

Formal interaction models specify causal relationships between *inputs*, *outputs*, and *internal states* of the AI system and users. In particular, a formal interaction model specifies "state updates": how the outputs (or decisions) of the AI system affect the internal state of each individual user, and how the outputs of each individual user (e.g., observed behaviors) affect the state of the AI system. A formal interaction model also specifies the relationship between the user's internal state and output (and similarly for the AI system). The key component of a formal interaction model is how the AI system outputs shape the internal state of users, and we thus focus on models in which the internal state of each user is *nontrivially affected* by the outputs of the AI system.

We also focus on formal interaction models which capture users at the *individual level*, taking complementary perspective from *population-level* approaches to mathematical modelling of AI-user interactions. Examples of population-level approaches are the framework of performative prediction (Perdomo et al., 2020) (which formalizes how a predictor can shape the distribution of users in supervised learning) and the framework of reinforcement learning (which abstracts *population-level dynamics* into an "environment" in which sequential decision-making occurs (Sutton & Barto, 2018)). Individual-level models specify granular behaviors, and they can be aggregated and analyzed at the population level to draw conclusions about societal impacts (see e.g. how individual-level strategic behavior results in population-level distribution shift (Perdomo et al., 2020)).

Returning to the content recommender system example, the AI system refers to the recommender system, and "users" can refer to either viewers or creators. The internal state of viewers captures viewer preferences or other behaviors, and the internal state of creators captures internal decisions about what content to create or whether to participate on the recommendation platform. The outputs of the recommender system are recommendations, the outputs of viewers are observed behaviors such as clicks, and the outputs of creators are uploaded content or participation outcomes. We describe formal interaction models for content recommendation in more detail in Section 3.

**Definition of a Formal Interaction Models as a Coupled Dynamical System.** To further distill the definition of a formal interaction model, we cast a formal interaction model as a *coupled dynamical system*. This formulation serves as a useful unifying language to describe many different mathematical models for

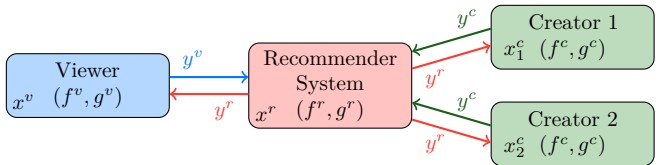

Figure 2: A formal interaction model for content recommendation as a coupled dynamical system between the recommender system $(x^r, y^r, f^r, g^r)$, viewers $(x^v, y^v, f^v, g^v)$, and creators $(x^c, y^c, f^c, g^c)$. The variable $x$ captures the internal state, the function $f$ captures the state update, the function $g$ captures the measurement equation, and the variable $y$ denotes outputs and inputs (Section 2.1 and Section 3).

user-AI interactions. We first provide a brief overview of the dynamical systems language and then cast formal interaction models within this language.[4]

Following Willems (1989), a (causal) dynamical system is a mathematical object describing the evolution of various quantities over discrete time steps $t \geq 0$ (Figure 1a). Its definition is grounded by observable quantities, which come in two types: inputs, denoted by $u$, come from a possibly unexplained environment; and outputs, denoted by $y$, come from the dynamical system model. It is often convenient to introduce an *internal state variable* $x$ which is internal and evolves over time $(x_t)_{t \geq 0}$. At its most general, a dynamical systems model specifies relationships between sequences of inputs $(u_t)_{t \geq 0}$ and sequences of outputs $(y_t)_{t \geq 0}$. The only universal restriction is that the model is *causal*, meaning that the output $y_t$ is not affected by the future inputs $u_k$ for $k$ greater than $t$.

A dynamical systems model can be formalized by an *internal state variable* $x$ (how the state is represented), a *state update* $f$ (how the state is affected by inputs) and a *measurement equation* $g$ (how the state affects outputs). The functions $f$ and $g$ recursively define the possible outputs of a system in terms of its inputs and an additional time-dependent *disturbance* variable $w_t$ which accounts for non-stationary, stochastic, or adversarial effects. More specifically, the internal state is updated according to $x_{t+1} = f(x_t, u_t, w_t)$ and the outputs are updated according to $y_t = g(x_t, u_t, w_t)$.

Using the dynamical systems language, *formal interaction models* can be expressed as a *coupled dynamical system*—i.e., multiple, interconnected dynamical systems—as shown in Figure 1b. Each user $i$'s behavior is captured by a dynamical systems model $(x^i, f^i, g^i)$, and the AI system's behavior is also captured by a dynamical systems model $(x^a, f^a, g^a)$. These dynamical systems are *coupled* in that the *outputs* $y_t^a$ of the AI's dynamical system are the *inputs* $u_t^i$ of each user's dynamical system, and the *outputs* $y_t^i$ of each user's dynamical system together form the *input* $u_{t+1}^a$ of the AI system's dynamical system. The key component of a formal interaction model is the state update $f^i$ for each user, which captures how the AI system shapes the user. To capture meaningful dynamics, this state update must *nontrivially* change with the output $y^a$.

For the special case of content recommendation, a formal interaction model is a coupled dynamical system between the recommender system, viewers, and creators (Figure 2). We defer an extensive discussion of content recommendation to Section 3, but provide a simplified example here to build intuition.

**Example 1** *Consider a model of user opinion in a video recommendation setting. For simplicity, consider videos representing opinions along a single axis, e.g. on plant-based vs. meat-based diets. Let $y^i \in \mathbb{R}_+$ denote the user watch time and $y^{a,i} \in \mathbb{R}$ denote the recommended video, parametrized by this single opinion axis.*

*Let the user's internal state $x^i \in \mathbb{R}$ be their own scalar opinion. Suppose that watchtime depends on the extent to which the user agrees with the video. Further suppose that the strength of user opinions tends to decay, but that they "assimilate" the opinions contained in the recommended videos. This can be formalized by the state update[5] $x_{t+1}^i = f^i(x_t^i, y_t^{a,i}) = 0.9 \cdot x_t^i + 0.1 \cdot y_t^{a,i}$ and the measurement equation*

---

[4]The dynamical systems language (Willems, 1989) is appealing for describing formal interaction models for the following reasons: it provides a clear formulation for cause and effect, it is well-suited to applications due to its focus on measurable quantities, and yet it retains sufficient generality to describe many different types of models.

[5]Note that if opinion were *independent* of videos (e.g., if $x_{t+1}^1 = 1$ at all time steps), then this model would *not* be counted as a formal interaction model (or alternatively, would be a "trivial" formal interaction model).

$y_t^i = g^i(x_t^i, y_t^{a,i}, w_t^i) = \sigma(y_t^{i,a} \cdot x_t^i + w_t^i)$ *where $w_t^i$ is noise accounting for exogenous factors and $\sigma$ is some activation function mapping to $[0, 1]$.*

*The recommender system's internal state is an estimate of user preferences based on observed watchtimes. Recommendations $y_t^{a,i}$ are chosen to maximize predicted watchtime. As a result, users watch videos that reinforce their existing opinions (so long as the recommender adequately learns from data). At a population level, this induces* polarization*, wherein user opinions are positive or negative but nonzero, e.g. in favor of plant or meat based diets rather than omnivorous.*

**Examples of Formal Interaction Models for User-AI Interactions.** Due to the widespread ability of AI systems to shape users, formal interaction models are starting to be developed across many domains.[6] For example, in *content recommendation*, a nascent body of work (which we survey in Section 3) proposes models for how content recommender systems can shape viewers and creators. As another example, formal interaction models in the *strategic classification* literature (e.g. Hardt et al., 2016; Kleinberg & Raghavan, 2019; Rosenfeld, 2024) capture how a decision rule incentivizes individual users to change their features to try to receive a positive prediction. Moreover, other works have proposed formal interaction models for how *fairness interventions* shape dynamics (e.g. Pagan et al., 2023; Reader et al., 2022; Liu et al., 2018), how *autonomous vehicles* shape driver behavior (e.g. Sadigh et al., 2016), and how a *foundation model* shapes the finetuning or prompting decisions of service providers (e.g. Jagadeesan et al., 2023; Laufer et al., 2023).

While these nascent literatures are promising, formal interaction models for user-AI systems are still limited in scope and have not been sufficiently leveraged in the design and evaluation of AI systems. This highlights the continued need to develop (and expand the scope of) formal interaction models for user-AI systems.

## 2.2   Use Cases of Formal Interaction Models

We argue that formal interaction models can play an instrumental role in *specifying* interactions for implementation, *monitoring* interactions through empirical analysis, *anticipating* societal impacts via counterfactual analysis, and *controlling* societal impacts via interventions. Our argument draws inspiration from Abebe et al. (2020) which outlines the roles of computer science for social change.

**Specify interactions for implementation.** Developing formal interaction models forces the research community to fully specify AI-user interactions, so that these models can be *implemented* to empirically capture feedback effects.

Implementing formal interaction models can improve the empirical evaluation and design of AI systems across many different learning paradigms. For example, in the context of *model-based RL*, a formal interaction model—in particular, each user's internal state $x^i$ and state update $f^i$—can be incorporated into the RL state representation and the transition probabilities between states. As another example, the quantities $x^i$ and $f^i$ of a formal interaction model can also be implemented in *environment simulators* to empirically capture how users are shaped by the actions taken by the AI systems. These environment simulators can be used to train and evaluate *model-free RL* (where the transitions probabilities no longer need to be specified ahead of time) and also to evaluate *supervised learning-based systems* (which do not explicitly account for interactions with users). As an example, in fair machine learning, D'Amour et al. (2020) design simulators to evaluate how feedback dynamics affect fairness in the context of bank loans, college admissions, and allocation of attention. In content recommendation, an emerging body of work by Rohde et al. (2018); Ie et al. (2019); Krauth et al. (2020); Mladenov et al. (2021); Gao et al. (2023); Deffayet et al. (2023) designs simulators to evaluate how recommender system performance is affected by interactions with users over time.[7]

**Monitor interactions through empirical analysis.** Developing formal interaction models can make it more tractable to monitor interaction patterns between AI systems and users in real-world deployments.

---

[6]Even well before the emergence of modern AI systems, formal interaction models were developed in classical decision-making systems, for example in mechanism and market design as well as in physical systems (Appendix A).

[7]These simulation frameworks could serve as useful testbed to implement and test the formal models that we survey in our case study (Section 3).

Formal interaction models can be leveraged to monitor the strength and character of interactions between AI systems and users in real-world environments. These mathematical models can surface relevant constructs to monitor and can also formalize these constructs as variables (e.g., finite-dimensional parameters) which can be estimated from data about real-world interactions. The estimated parameters can be assessed to determine whether the interactions are strong (or weak) in a given environment and to shed light on other qualitative properties (such as which users are most affected).[8] As an example, in strategic classification, Björkegren et al. (2020) measure manipulation costs in a large field experiment in Kenya, which provides insight into the strength and character of strategic interactions. As another example, in content recommender systems, Milli et al. (2021) measure how well different engagement signals (and combinations of signals) capture value, and Aridor et al. (2023) measure the causal effect of recommendations on viewer beliefs about content quality.

Even observing that a formal interaction model does *not* capture real-world interactions itself provides valuable information. In particular, falsifying a model demonstrates that either the structure of interactions might be different than anticipated (which motivates adjusting the model) or that the model might miss important interaction patterns (which motivates augmenting or expanding the model). As an example, in strategic classification, Jagadeesan et al. (2021) observed that the standard agent models fail to capture distribution shifts observed in practice and also have limited prescriptive value. As another example, Leqi et al. (2023) tested the fixed reward distribution assumption in multi-armed-bandit-based recommender systems and found it to be violated. In such cases, falsifying existing models through empirical analysis can encourage the community to refine models to better capture empirical realities.

**Anticipate societal impacts via counterfactual analysis.** Developing formal interaction models enables the community to anticipate societal impacts driven by AI-user interactions in counterfactual environments (e.g., where AI algorithm changes or user interactions change).

Formal interaction models can be leveraged to characterize when societal impacts occur in existing domains. For example, analyses of formal interaction models (via theoretical analyses or simulations) can reveal how the algorithm parameters and the broader environment together influence the strength and nature of the societal impact. These analyses can enable the community to anticipate the impacts of AI-user interactions in counterfactual environments which could arise with changes to the algorithm or users. As an example of an empirical analysis, in fair machine learning, D'Amour et al. (2020) uses the simulations to compare and assess the long-term fairness of different algorithms; as an example of a theoretical analysis, in strategic classification, Kleinberg & Raghavan (2019) characterize how the scoring rule shapes how much effort agents invest in improvement versus gaming.

Formal interaction models can help forecast what societal impacts will occur in emerging domains. This is particularly important when new technologies are introduced into the world. We expect that some principles underlying formal interaction models in existing domains can be translated to new domains, which could help anticipate potential societal impacts of new technologies before they occur. For example, to forecast behaviors about emerging LLM-based ecosystems, Jagadeesan et al. (2023); Laufer et al. (2023) develop formal interaction models which borrow inspiration from industrial organization and bargaining games.

**Control societal impacts via interventions.** Developing formal interaction models can help control societal impacts through either through improving algorithm design or highlighting avenues for broader sociotechnical interventions (such as public policy).

Formal interaction models can surface new metrics that can be optimized by supervised learning or RL-based algorithms to control societal impacts. Certain societal impacts may be directly measurable by a single parameter of the mathematical model, whereas other societal impacts may arise as complex combination of multiple parameters. We expect that crafting new metrics which capture these societal impacts could benefit the design of objectives, which is classically an ad-hoc process that requires trial-and-error. As an example, for the Twitter platform, Milli et al. (2021) propose metrics to measure value rather than engagement. As a broader research agenda, Boutilier et al. (2023) propose leveraging tools from mechanism design to recommender systems design which optimize downstream user welfare and ecosystem health.

---

[8]Parameter estimation is a common use-case of formal interaction models in econometrics and empirical industrial organization.

Formal interaction models can also help highlight avenues for broader sociotechnical interventions. It may be intractable to control certain societal impacts purely via algorithm design, or the platform may not be incentivized to take responsibility. In such cases, formal interaction models may inform how to disrupt AI-user interactions with non-algorithmic design choices (e.g., UI choices) or with public policy. For example, seminal work in fair machine learning (e.g. Kleinberg et al., 2017; Chouldechova, 2017) used mathematical models (though not formal interaction models) to show inherent tradeoffs between different fairness definitions for any scoring rule; these works illuminates the boundaries of what is achievable with technical interventions (Abebe et al., 2020). We expect that the formal interaction models can similarly be leveraged to inform the limits of technical interventions and ultimately inform the design of broader sociotechnical interventions.

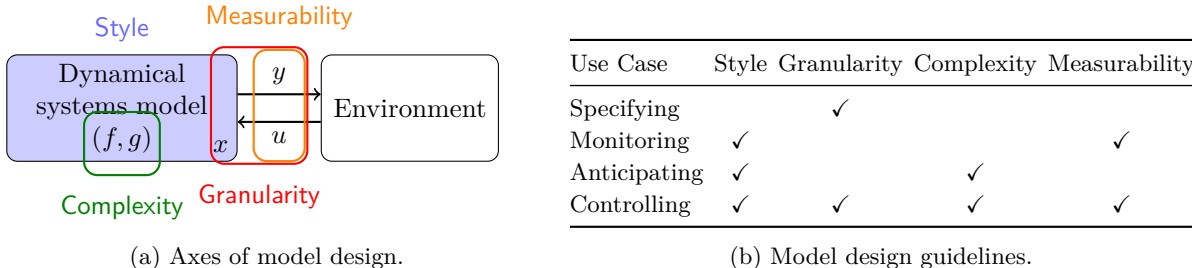

| Use Case | Style | Granularity | Complexity | Measurability |
|---|---|---|---|---|
| Specifying | | ✓ | | |
| Monitoring | ✓ | | | ✓ |
| Anticipating | ✓ | | ✓ | |
| Controlling | ✓ | ✓ | ✓ | ✓ |

(a) Axes of model design.   (b) Model design guidelines.

Figure 3: *Left:* A breakdown of which design axes correspond to which components of the formal model. *Right:* Guidelines for model design along different axes based on intended use case. Checkmarks indicate which axes are especially important for each use case.

## 2.3 Axis of Varying Models: Guidelines for Design Choices

Formal interaction models can vary along several design axes—*style*, *granularity*, *mathematical complexity*, and *measurability*—which creates a vast design space of possible models. There is no "one-size-fits-all" approach: which approach is most suitable depends on the particular use case. In this section, we grapple with how model design requires careful consideration along style, granularity, mathematical complexity, and measurability. We outline each design axis, and offer guidance on the design decisions tailored to each use-case. Figure 3 summarizes the design axes and the guidance on design decisions.

**Axis I: Style.** The style of a formal interaction model depends on the discipline(s) on which model design is based. For example, nonlinear dynamics-based models often have roots in control theory, reward models are often based on psychology and behavioral economics, and game-theory-based models often incorporate perspectives from microeconomics and industrial organization. The model style particularly affects how the user internal state $x^i$ is represented and how the state update $f^i$ is specified. For example, while these quantities are explicitly defined in nonlinear dynamics-based models, they are only implicit in the strategy and utility function defined by game theory-based models. Example 1 is a (linear) dynamics-based model.

**Axis II: Granularity.** The granularity of a model describes the level of detail to which a model is specified, i.e. the specificity of the relationships between inputs, outputs, and states. Examples of highly granular models are fully parametric models (e.g. linear models). At the other extreme, low-granularity models might only specify only inputs and outputs and not incorporate any further structure. At the middle of the spectrum are models with structure beyond their input and output, but do not fully specify functional forms. This includes graphical models (such as those used in casual modelling) which illustrate dependencies, but either do not specify any functional relationships or place an additive structure which allows for flexibility in each component. Example 1 is a granular model, since the relationships are exactly specified with given parameters.

**Axis III: Mathematical Complexity.** The complexity of a model captures the overall mathematical intricacy of the model. Key factors which affect model complexity are the intricacy of the state update $f^i$ (for how AI outputs shape the internal state of users) and the intricacy of the measurement equation $g^i$ (which determines user outputs). Complex models often have a comprehensive set of variables and a flexible

functional form with the goal of capturing all factors of interest. At the other extreme, simpler models are usually more interpretable and focus only on first-order factors. Example 1 is a low complexity model with scalar variables and one dimensional functions.

**Axis IV: Measurability.** The measurability of a model captures how feasible it is to measure the mathematical constructs of the formal interaction model with real-world observable data. Examples of highly measurable models include parametric models based on observable constructs such as watch time and clicks. On the other hand, some models incorporate constructs (e.g., the psychological state of a user) that may be naturally hard or impossible to observe. Nonetheless, the model still may be partially measurable if the model is identifiable and observable data can be used to infer the underlying parameters. However, the model may not be measurable if the functional form of a model is unidentifiable, in which case observable data is insufficient to infer model parameters. When a formal interaction model contains unobservable and unidentifiable constructs, this can serve as a call to action for measuring additional variables or re-conceptualizing design of the system (Jacobs & Wallach, 2021). Example 1 is measurable due to the simple specified relationship between observable watchtime and unobserved opinion.

**Guidelines for Design Choices.** Given that the four axes described above—style, granularity, mathematical complexity, and measurability—create a vast design space for formal interaction models, we reflect on best practices for designing a formal interaction model. One challenge is that the most appropriate design choices highly depend on the intended use-case of the model, and there is thus no one-size-fits all approach for model design. We thus offer high-level guidelines tailored to each use-case, highlighting which axes we believe are most important to consider for each use-case.

- *Specifying interactions*: When specifying interactions for the purpose of designing simulators, we view granularity as the most important axis, and complexity as relevant. In particular, to empirically capture feedback effects, it is necessary to design a *granular model* with sufficient *complexity* to fully specify how the users react to the AI system outputs.

- *Monitoring interactions*: When monitoring AI-user interactions, designing a *measurable model* is key. In particular, the constructs that are monitored need to be measurable from observable data, both to determine the strength or character of AI-user interactions and to validate (or falsify) the model. Furthermore, *model style* should be tailored to the aspects of interactions that one aims to monitor. For example, to monitor strategic behaviors of users, a game theory-based model would be helpful; to monitor the preference shifts of a user, a non-linear dynamics based models may be more appropriate.

- *Anticipate societal impacts*: When anticipating societal impacts, designing a *mathematically simple model* can help capture the salient aspects of the ecosystem and uncover the drivers of a societal impact. Relatedly, *model style* should be tailored to which factors that drive the societal impact of interest. For example, nonlinear dynamics-based models are particularly appropriate for capturing convergence and evolution over time. As another example, reward models in online decision-making settings (e.g., multi-armed bandits) are well-suited to capturing how decisions impact perceived rewards. As yet another example, game-theory models are helpful for capturing the equilibrium state when many users compete with each other.

- *Control societal impacts*: When controlling societal impacts via algorithm design, the *choices along multiple axes* impact the practicality and robustness of an algorithmic intervention. For example, measurability affects whether the construct of interests can be optimized from the observable data available to the AI system. Furthermore, mathematical simplicity affects the tractability of obtaining provable guarantees. Finally, the granularity affects the robustness of interventions, since models which specify exact functional forms can be subject to model misspecifications.

Taking these guidelines into account, Example 1 is perhaps best suited to *anticipating societal impacts* related to polarization over time, as it is a simple dynamics-based model. In contrast, its complexity is likely too low to be meaningfully used in *specifying*, *monitoring*, or *controlling*, since opinion shifts may depend on interactions between many factors, rather than a single scalar value.

|  | Motivating Phenomenon | Style | Use Case |
|---|---|---|---|
| Viewer Preference | echo chambers, polarization, boredom | dynamics, bandit | anticipating, controlling |
| Creator Content | diversity, clickbait | game theory, bandit | anticipating, controlling |
| Viewer Participation | over-consumption, retention | behavioral econ, bandit | anticipating, controlling |
| Creator Participation | retention | game theory, bandit | anticipating, controlling |
| Other | miscellaneous | game theory, behavioral econ | anticipating, controlling |

Table 1: Summary of formal interaction models in content recommendation, addressing viewer preferences, creator content, participation, and other behaviors.

## 3 Case Study: Content Recommendation

In this section, we return to our motivating example of content recommendation (Section 2.1) as a case study. Our goal is to critically examine an emerging line of work which grapples with the ways that content recommender systems shape viewers and creators. To compare these models and place them on a common ground, we cast these models within the unifying framework of coupled dynamical systems (Section 2.1; Figure 2). We investigate how the formal interaction models proposed in this line of work capture the use-cases outlined in Section 2.2. We also consider model design along the axes outlined in Section 2.3. Table 1 summarizes our analysis.

We carry out this analysis for formal interaction models where *viewer or creator internal states are shaped by the recommender system.* This includes viewer preferences (Section 3.1), creator content (Section 3.2), and other behaviors (Section 3.3). For each model, we focus on how the AI system shapes users: in the dynamical systems language, this is the state $x^i$ and state update $f^i$ for each user. Note that in this analysis, we reflect on lines of work as a whole, rather than surveying specific models in individual papers. The collection of papers included in the analysis was generated as follows: first, a small set of key papers was identified; then, it was expanded following the citation graph; finally, only papers defining a *non-trivial* formal interaction model where included. The collection includes papers published through early 2024.

Our analysis uncovers several limitations of this nascent literature and reveals interesting opportunities for future work (Section 3.4). First, in terms of use-cases, we find that most of the focus has been on anticipating societal impacts and controlling societal impacts; this highlights the need for future work focused on specifying interactions or monitoring interactions. Second, in terms of model design, we find that the models are fairly homogeneous in style within each interaction type, and that models of viewer interactions and creator interactions have been largely separate; this highlights an opportunity to broaden the scope of these models.

Recall that we do not consider dynamics models where the user state does not change (e.g. Chaney et al., 2018; Mansoury et al., 2020) to be formal interaction models (Section 2.1). Such models do still induce feedback loops because viewers provide the recommender system with feedback about their utility from recommended content, and the recommender system uses viewer feedback to estimate viewer utilities. However, even though the recommender system's embeddings of viewer preferences can change, the viewers' internal states (i.e., preferences) are not changing. These models are thus outside of the scope of our case study.

### 3.1 Viewer Preferences

One line of work investigates how recommendations affect viewer preferences and opinions. In these papers, the primary setting is an individual viewer consuming content and responding with feedback such as watch time, clicks, or likes. These works focus on how different recommendation algorithms shape a viewer's internal state, how such state changes result in broader societal impacts, and how a platform can design algorithms to control the negative societal impacts. Below, we present an overview and critical examination of these works, and we defer detailed mathematical definitions to Appendix B.

**Societal impacts.** One motivation for this line of work is the unintended consequences of personalized recommendation. For example, societal impacts—such as echo chambers, polarization, biased assimilation, manipulation (e.g. Lu et al., 2014; Dean & Morgenstern, 2022)—can emerge from engagement-based recommendations shaping viewer preferences. Moreover, when viewers get bored of content, this can lead to broader

forms of misalignment between the goal of recommender systems and the intended goal of recommender systems and what they actually optimize.

**Style of the models.** Two dominant styles of models are latent factor-based models (e.g. Dean & Morgenstern, 2022) and multi-armed bandits models (e.g. Kleinberg & Immorlica, 2018). The latent factor models in the literature capture viewer preference shifts through viewer embedding dynamics, typically taking the perspective that recommendations shape how viewers see the world. The multi-armed bandit models capture preference updates in terms of changes in the reward distributions, often taking the perspective that viewers' preference shifts occur naturally as viewers get bored of content.

**Overview of models.** While these models differ in the type of preference shifts they examine, there are commonalities among the state definitions, state updates, and measurement functions. The *state* $x^v$ represents viewer preferences or opinions that can be represented as vectors or scalars that are discrete (e.g. Jiang et al., 2019; Agarwal & Brown, 2023) or continuous (e.g. Dean & Morgenstern, 2022; Curmei et al., 2022) observable (e.g. Levine et al., 2017; Kleinberg & Immorlica, 2018) or latent (e.g. Leqi et al., 2021; Ben-Porat et al., 2022). The *measurable behaviors* $y^v$ of the viewers are the ratings or clicks. The *state update* $f^v$ models viewer preference changes due to recommendations, as a result of exposure to a type of content (e.g. Kalimeris et al., 2021; Evans & Kasirzadeh, 2021), viewers' selections (e.g. Brown & Agarwal, 2022; Agarwal & Brown, 2023), or their measured behavior (e.g. Dean & Morgenstern, 2022). Viewer preferences can shift towards the recommended content (e.g. Carroll et al., 2022; Krueger et al., 2020) or away from recommendations (e.g. Lu et al., 2014). The state update can also capture the viewer's memory accumulation of (the quality of) past recommended content (e.g. Curmei et al., 2022), which can result in viewer boredom (e.g. Levine et al., 2017; Pike-Burke & Grunewalder, 2019) or loyalty (e.g. Krueger et al., 2020). Finally, $f^v$ may also have a periodic structure, capturing viewers' seasonal preference shifts (e.g. Laforgue et al., 2022; Foussoul et al., 2023).

**Use-cases of these models.** In terms of use-cases, this line of work has focused on anticipating and controlling societal impacts. The nonlinear-dynamics models are utilized to identify failure modes of existing recommender systems (e.g., biased assimilation, echo chamber, manipulation) and propose adjustments to existing algorithms to address these issues (e.g. Carroll et al., 2022; Evans & Kasirzadeh, 2021). Meanwhile, the bandit-style models are used to develop recommendation algorithms attuned to impacts on viewers' transient preference (e.g. Foussoul et al., 2023; Khosravi et al., 2023).

## 3.2 Creator Content

Shifting focus from viewers to creators, a different line of work investigates how a content recommender system shapes (and is shaped by) the *landscape of content* available on the platform. The high-level mechanism is that content creators compete to appear in recommendations: this means that the recommendation functions shapes creator incentives around what content to create, which affects the composition of the content landscape. Below, we present an overview and critical examination of these works, and we defer detailed mathematical definitions to Appendix C.

**Societal impacts.** One motivation for this line of work is societal impacts emerging from a recommender system shaping the content landscape. For example, shifts in the content landscape affect viewer satisfaction and platform metrics and contribute to broader phenomena such as polarization and addiction. Thus far, this line of work has focused on the *quality* and *diversity* of content and resulting impact on recommendations: e.g., creator effort in quality (e.g. Ghosh & McAfee, 2011), content specialization (e.g. Jagadeesan et al., 2022; Eilat & Rosenfeld, 2023), creators relying on gaming tricks such as clickbait (e.g. Buening et al., 2023; Immorlica et al., 2024), and implications for engagement and welfare (e.g. Yao et al., 2023b).

**Style of the models.** These models take the perspective of *game theory*, formalizing how creators—who compete to win recommendations—strategically design their content to appear in as many recommendations as possible. As is common in game theoretic models, the focus is on the equilibria (which corresponds to *fixed points* of the underlying dynamical system). A handful of works have introduced dynamics over time by studying convergence of best-response and better-response dynamics (e.g. Ben-Porat et al., 2020; Hron et al., 2022) and by incorporating the platform learning over multiple time steps (e.g. Ghosh & Hummel, 2013).

**Overview of models.** While these models take the perspective of game-theory, we cast these models within the dynamical systems framework to enable better comparison with the interaction patterns for viewer preferences in Section 3.1. The *state* $x_j^c$ of each creator $j$ captures the creator's internal decision about what content to create and the *output* $y_j^c$ captures the content that the creator uploads. Across different models, the state space ranges from a finite set (e.g. Ben-Porat & Tennenholtz, 2018), to $\mathbb{R}_{\geq 0}$ (e.g. Ghosh & McAfee, 2011), to $\mathbb{R}^D$ (e.g. Hron et al., 2022; Jagadeesan et al., 2022), to more abstract sets (e.g. Yao et al., 2023b). The *state update* $f^c$ captures the creator's best-response to their utility function. The creator utility function captures the creator's reward from recommendations—which ranges from exposure (e.g. Ben-Porat & Tennenholtz, 2018) to engagement (e.g. Yao et al., 2023a) to abstract functions (e.g. Yao et al., 2023b)—and also sometimes captures costs of production (e.g. Ghosh & McAfee, 2011). Here, the recommendation function $g^r$ implicitly shapes the creator's reward by affecting recommendations, and different papers study different recommendation functions.

**Use-cases of these models.** In terms of use-cases, this line of work has focused on *anticipating* and *controlling societal impacts.* For anticipating societal impacts, some works theoretically analyze how the engagement and welfare of standard supervised learning and bandit algorithms can degrade under creator competition (e.g. Ghosh & Hummel, 2013; Yao et al., 2023a; Hu et al., 2023); other works characterize when societal impacts (e.g., creator specialization) occur as a function of the recommendation function and marketplace specifics (e.g. Jagadeesan et al., 2022). For controlling societal impacts, some works consider the endogeneity of the content landscape and design recommender system objectives and learning algorithms which optimize for downstream engagement and welfare (e.g. Buening et al., 2023).

### 3.3   Other Behaviors

More broadly, recommender systems can also shape other aspects of users beyond viewer preferences and creator content. We briefly summarize a few of these aspects below.

**Viewer Participation.** One line of work studies how viewer participation is influenced by recommender system algorithms.

- *Behavior modelled.* These models capture how viewers may depart the system (e.g. Ben-Porat et al., 2022) or may regret their time on the platform (e.g. Kleinberg et al., 2022).

- *Overview of models.* The state $x^v$ captures whether the viewer stays on the platform or has departed (Huttenlocher et al., 2024; Ben-Porat et al., 2022); or it may represent one of the two consumption processes of a viewer (Kleinberg et al., 2022). The state update $f^v$ determines can be specified by a fixed leaving probability (Ben-Porat et al., 2022); or it can be determined by the quality of recommendation content at the previous time step (Huttenlocher et al., 2024; Kleinberg et al., 2022). Recommender systems are modeled as bandit or matching algorithms.

- *Use-cases.* Formal models developed in this line of work are used to anticipate and control societal impacts: analyzing how recommender systems may affect viewers to stay longer than they had intended and designing algorithms that account for viewer departure behaviors.

**Creator participation.** Another line of work examines how the recommender system can shape whether creators stay on or leave the platform.

- *Behavior modelled.* These models formalize how creators need to achieve sufficient recommendation exposure or engagement with their content to stay on the platform (e.g. Mladenov et al., 2020).

- *Overview of models.* The state $x^c$ incorporates the creator's (probabilistic) internal decision about whether to stay on the platform and the the creator's cumulative level of exposure (e.g. Mladenov et al., 2020; Ben-Porat & Torkan, 2023). (In some models, the creator state also includes creator decisions about content quality (Ghosh & McAfee, 2011; Ghosh & Hummel, 2013).) The state update $f^c$ captures that the creator will leave the platform if they do not receive sufficient exposure or engagement over a certain

period of time. Across the different spaces, the recommender systems range from a full-information (e.g. Mladenov et al., 2020) to a bandit algorithm (e.g. Ghosh & Hummel, 2013).

- *Use-cases.* This line of work has primarily focused on anticipating and controlling societal impacts: in particular, determining when a recommendation algorithm incentivizes creators to leave the platform, and designing recommendations algorithms which retain creators.

**Other decisions made by viewers** Another line of work examines how the recommender system can shape broader forms of viewer decisions beyond participation.

- *Behavior modelled.* These models formalize viewers deciding to not follow recommendations (e.g. Mansour et al., 2020), strategically deciding what content to engage with to curate their feed (Haupt et al., 2023; Cen et al., 2023), and switching between different behavioral modes (e.g. Kleinberg et al., 2022). The style of model ranges from game-theoretic to behavioral economics-inspired.

- *Overview of models.* The state $x^v$ in these models captures viewer internal decisions, ranging from a vector-valued decision about which content to engage with (Haupt et al., 2023; Cen et al., 2023), to a binary decision about whether to follow recommendations (Mansour et al., 2020), to a behavioral model controlled by System 1 or System 2 (Kleinberg et al., 2022). The state update function $f^v$ ranges from a best-response to shape the trajectory of future recommendations (Haupt et al., 2023; Cen et al., 2023) to implicit probabilistic updates depending on the specifics of the content consumed (Kleinberg et al., 2022).

- *Use-cases.* This line of work has also primarily focused on anticipating and controlling societal impacts.

### 3.4 Discussion and Opportunities for Future Work

**Use cases.** Returning to the use cases outlined in Section 2.2, our analysis revealed that most of the formal interaction models for content recommendation have focused on anticipating and controlling societal impacts. Thus far, this line of work has devoted limited attention to specifying interactions (e.g., for designing comprehensive and realistic simulators) or monitoring interactions (e.g., for assessing the characteristics of real-world interactions). Part of this results from the disconnect between the community that develops formal models and the community that empirically builds simulators or develops recommender systems. To broaden the use cases of formal models, we believe that a centralized platform—that utilizes the common language of coupled dynamical systems—would be helpful for researchers to consolidate formal models and for practitioners to integrate these models into the design and evaluation of recommender systems.

**Enriching the models for each type of interaction.** Our analysis revealed that within each interaction type, the models are fairly homogeneous in style. For example, the viewer preference literature has primarily studied nonlinear dynamics-based models, and the creator content literature has primarily studied game-theory based models. This homogeneity in model style has implicitly limited the focus of these models: the viewer preferences models has primarily focused on the dynamics between the recommender system and an individual viewer, whereas the creator content models has primarily focused on how competition between multiple creators affects behavior at equilibrium. An important direction for future work is to enrich the style of models for viewer interactions and creator interactions. This includes investigating how interactions between *multiple viewers* could affect preferences, for example, by incorporating network structure (e.g. Crandall et al., 2008). This also includes incorporating how *(nonlinear) dynamics of creators* over time affect the content landscape as well as incorporating how *recommendation biases* and *prices* affect market concentration (e.g. Calvano et al., 2023; Castellini et al., 2023).

**Interplay between multiple interaction types.** Our analysis also revealed that the models of viewer interactions and creator interactions are largely separate. However, in reality, we expect that the interplay between viewer interactions and creator interactions drives broader societal impacts such as polarization or addiction. An important direction for future work is thus to integrate formal interaction models of viewers and formal interaction models of creators into a single coupled dynamical system. Recent work (Huttenlocher et al., 2024) takes a step in this direction and simultaneously considers both viewer and creator participation

decisions. We encourage future work to incorporate richer forms of interplay between creator interactions and viewer interactions, which we expect will involve analyzing a complex coupled dynamical system (Figure 2).

## 4 Conclusion and Discussion

In this survey paper, we argued for the development of *formal interaction models* for how AI systems and users shape one another. We defined a formal interaction model as any coupled dynamical system between an AI system and individual users which captures how the AI system shapes the internal state of users. We outlined how formal interaction models can be leveraged to (1) specify interactions for implementation, (2) monitor interactions through empirical analysis, (3) anticipate societal impacts via counterfactual analysis, and (4) control societal impacts via interventions. We also discussed how model design needs to be tailored to the intended use-case, highlighting the role of model style, granularity, mathematical complexity, and measurability. Finally, using recommender systems as a case study, we critically examined the nascent literature of formal interaction models (e.g., of viewer preference shifts, content creator incentives, and other user behaviors). Using the dynamical systems language to place these models onto a common ground, we identified limitations in the use-cases and scope of these models, and uncovered opportunities for future work.

We expect that formal interaction models—if appropriately leveraged—can significantly improve AI system design and evaluation in *any* domain where AI systems and users interact with each other. This includes *content recommender systems*, which implicitly shape viewer preferences, the content landscape, and many other user behaviors (Section 3). This also includes *user-facing decision-support systems* (integrated into hiring pipelines (Commission et al., 2022), credit score assessment (Sadok et al., 2022), and resource allocation (Obermeyer et al., 2019)) which can lead to strategic behavior (Rosenfeld, 2024) and affect access to opportunities and fairness objectives (Reader et al., 2022; Pagan et al., 2023). As yet another example, this includes *autonomous vehicles*, which interact with human drivers and can affect society-level traffic patterns and safety of pedestrians (Sadigh et al., 2016). In the coming years, we expect that this will also include *language models*, which are being augmented with external tools and integrated into broader ecosystems, creating new forms of AI-user interactions (Pan et al., 2024).

As AI systems rapidly enter into more societal domains, the need for formal interaction models is increasing, but the research community has not prioritized developing and leveraging these models. We call for the research community to realize the promise of formal interaction models and integrate these models into the design and evaluation of AI systems.

## 5 Acknowledgments

We thank Micah Carroll for many insights into preference dynamics models. We also thank Mihaela Curmei, Alex Pan, and Abhishek Shetty for valuable feedback on a draft of this paper. This work was supported in part by NSF CCF 2312774, NSF OAC-2311521, an Open Philanthropy AI fellowship, a LinkedIn Research Award, and a gift from Wayfair.

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

## A  Formal Interaction Models in Classical Decision-Making Systems

Formal interaction models have historically informed the design of classical decision-making systems, well before the emergence of modern AI systems. We summarize two "success stories":

- In *mechanism and market design*, a clear understanding of how a mechanism impacts participant incentives—via a formal interaction model between the mechanism and participants—has been crucial to designing efficient marketplaces. For example, in the *medical residency match*, formal interaction models (capturing how the matching mechanism impacts the incentives of residents and hospitals) informed the design of NRMP match and led to a Nobel Prize in Economics (Roth, 2018). Formal interaction models have also informed market design for *kidney exchange* and *school choice* (Roth, 2018).

- In *physical systems*, a clear understanding of how the system affects the physical world—via a formal interaction model, more commonly called a "dynamics" model—has been crucial for successful decision-making and control. For example, decision-making in *aerospace systems* involves navigating and controlling aircrafts and often leverages models of aerodynamics, orbital mechanics, among many others. Control systems frequently leverage these models to calculate optimal paths, manage fuel efficiency, and ensure the stability of the vehicle under various physical conditions. For example, dynamics models have recently been leveraged by SpaceX to develop reusable orbital launch systems.[9]

Given the success of formal interaction models in classical decision-making systems, we expect that formal interaction models can similarly drive positive societal impact for AI systems as well.

## B  Additional details for Preference Shifts in Section 3.1

We discuss each paper, describe how the states and measurement variables are defined, and provide the formal models of state updates and measurement equations. In these models, $S^{d-1}$ denotes the sphere in $d$ dimensions.

We begin by discussing papers that are motivated by phenomena like echo chambers and polarization.

- Lu et al. (2014) define for each viewer $i$ the preference state $x_{i,t}^v \in \mathbb{R}^d$, the recommended content $y_{i,t}^r \in \mathbb{R}^d$, and the measured rating $y_{i,t}^v \in \mathbb{R}$ for some latent dimension $d$. The dynamics capture three possible behaviors: stationary, attraction, and aversation:

$$x_{i,t+1}^v \begin{cases} \sim \mu_{i,0} & \text{w.p. } \alpha_1 \\ = \sum_{\tau=1}^t w_{t-\tau} y_{i,\tau}^r & \text{w.p. } \alpha_2 \\ = -\sum_{\tau=1}^t w_{t-\tau} y_{i,\tau}^r & \text{w.p. } \alpha_3 \end{cases} \quad \mathbb{E}[y_{i,t}^v] = \langle x_{i,t}^v, y_{i,t}^r \rangle$$

- Curmei et al. (2022) define for each viewer $i$ the preference state $x_{i,t}^v \in \mathbb{R}^d$, the recommended content $y_{i,t}^r \in \mathbb{R}^d$, and the measured rating $y_{i,t}^v \in \mathbb{R}$ for some latent dimension $d$. The define dynamics for an effect "mere exposure"

$$x_{i,t+1}^v = (1-\alpha)x_{i,t}^v + \alpha y_{i,t}^r \quad \mathbb{E}[y_{i,t}^v] = \langle x_{i,t}^v, y_{i,t}^r \rangle \,.$$

They propose an additional model of "operant conditioning" with state variable augmented to include a memory variable $x_{i,t}^v = (p_{i,t}, m_{i,t}) \in \mathbb{R}^d \times \mathbb{R}$

$$\begin{bmatrix} m_{i,t+1} \\ p_{i,t+1} \end{bmatrix} = \begin{bmatrix} \delta(m_{i,t} + (p_{i,t}^v)^\top y_{i,t}^r) \\ (1-\alpha|s_{i,t}|)p_{i,t} + \alpha_1 s_{i,t} y_{i,t}^r \end{bmatrix}, \quad s_{i,t} = \arctan\left( \frac{1}{\sum_{\tau=1}^{t-1} \delta^\tau} m_t - (p_{i,t})^\top y_{i,t}^r \right), \quad \mathbb{E}[y_{i,t}^v] = \langle x_{i,t}^v, y_{i,t}^r \rangle$$

- Dean & Morgenstern (2022) define for each viewer $i$ the preference state $x_{i,t}^v \in \mathcal{S}^{d-1}$, the recommended content $y_{i,t}^r \in \mathcal{S}^{d-1}$, and the measured rating $y_{i,t}^v \in \mathbb{R}$ for some latent dimension $d$. The dynamics capture "biased assimilation," with

$$x_{i,t+1}^v \propto x_{i,t}^v + \eta_t \langle x_{i,t}^v, y_{i,t}^r \rangle \cdot y_{i,t}^r, \quad \mathbb{E}[y_{i,t}^v] = \langle x_{i,t}^v, y_{i,t}^r \rangle \,.$$

---

[9]See https://en.wikipedia.org/wiki/SpaceX_reusable_launch_system_development_program.

- Kalimeris et al. (2021) define for each viewer $i$ the recommendation including both content and predicted score $y^r_{i,t} = (s_{i,t}, c_{i,t}) \in \mathbb{R} \times \mathcal{C}$ and the measured click $y^v_{i,t} \in \{0, 1\}$. The viewer state $x^v_{i,t} \in [0, 1]$ is memoryless and determined by the predicted score:

$$x^v_{i,t} = \begin{cases} \sigma(s_{i,t}) + (1 - \sigma(s_{i,t})\gamma r(s_{i,t})) & s_{i,t} > 0 \\ \sigma(s_{i,t})(1 + \gamma r(s_{i,t})) & s_{i,t} \leq 0 \end{cases}, \quad \mathbb{E}[y^v_{i,t}] = x^v_{i,t}$$

- Jiang et al. (2019) define for each viewer $i$ the preference state $x^v_{i,t} \in \mathbb{R}^p$ the recommended length $k$ slate of content $y^r_{i,t} \in [p]^k$, and the measured binary feedback $y^v_{i,t} \in \{0, 1\}^k$ for $p$ discrete piece of content. They study a class of dynamics models wherein the preference for an item increases when it is clicked, and decreases when it is recommended but not clicked:

$$\forall\, j \in y^r_{i,t}, \quad \begin{cases} x^v_{i,t+1}[j] > x^v_{i,t}[j] & y^v_{i,t}[j] = 1 \\ x^v_{i,t+1}[j] < x^v_{i,t}[j] & y^v_{i,t}[j] = 0 \end{cases} \quad \mathbb{E}[y^v_{i,t}[j]] \text{ is increasing in } x^v_{i,j}[j]$$

- Agarwal & Brown (2023) define for each viewer $i$ the preference state $x^v_{i,t} \in \Delta(p)$, the recommended length $k$ slate of content $y^r_{i,t} \in [p]^k$, and the clicked item $y^v_{i,t} \in y^r_{i,t}$ for $p$ discrete piece of content. They study a class of dynamics models:

$$x^v_{i,t+1} = \frac{1}{\sum_{\tau=0}^{t} \gamma^\tau} (\gamma x^v_{i,t} + e_{y^v_{i,t}}) \quad y^v_{i,t} = c \text{ w.p. } \propto f^c(x^v_{i,t})$$

Brown & Agarwal (2022) study a similar model for the case that $\gamma = 1$.

- Krueger et al. (2020) define a global "loyalty" state $x^v_{g,t} \in \mathbb{R}^m$, for each viewer $i$ the preference state $x^v_{i,t} \in \mathbb{R}^p$, the recommended content $y^r_{i,t} \in [p]$, and the consumed content $y^v_{i,t} \in [p]$. A single viewer is active at each timestep with $i_t \sim \text{softmax}(x^v_{g,t})$. Then for this viewer $i = i_t$, the loyalty and preference update, and the viewer selects a content independent of the top recommendation.

$$x^c_{g,t+1}[i] = x^c_{g,t}[i] + \alpha_1 x^v_{i,t}[y^r_{i,t}], \quad x^v_{i,t+1} = \frac{x^v_{i,t} + \alpha_1 e_{y^r_{i,t}}}{\|x^v_{i,t} + \alpha_2 e_{y^r_{i,t}}\|_2}, \quad y^v_{i_t,t} \sim \text{softmax}(x^v_{i_t,t})$$

- Carroll et al. (2022) define for each viewer $i$ the preference state $x^v_{i,t} \in \mathbb{R}^d$, the recommended content $y^r_{i,t} \in \Delta(\mathcal{C})$ for a fixed content set $\mathcal{C} = \{c_1, \ldots, c_p\} \subset \mathbb{R}^d$, and viewer selection behavior $y^v_{i,t} \in \mathcal{C}$. The state update is influenced by a viewer belief over the future available content

$$x^v_{i,t} = x \text{ w.p. } \propto \exp(\beta_2(\lambda \bar{c}^\top x^v_{i,t} + (1-\lambda)\bar{c}^\top p)), \quad \bar{c} = \frac{\sum_{c \in \mathcal{C}} (y^r_{i,t}[c])^3 c}{\sum_{c \in \mathcal{C}} (y^r_{i,t}[c])^3}, \quad y^v_{i,t} = c \text{ w.p. } \propto y^r_{i,t}[c] \exp(\beta_1 x^{v\top}_{i,t} c)$$

- Evans & Kasirzadeh (2021) define for each viewer $i$ the belief state $x^v_{i,t} \in [0, 1]^3$, recommended content type $y^r_{i,t} \in \{1, 2, 3\}$, and viewer response $y^v_{i,t} \in \{0, 1\}$. For either $j, j' = 1, 3$ (when $x^v_{i,t}[1]$ is largest element) or $j, j' = 3, 1$ (when $x^v_{i,t}[3]$ is largest element), $x^v_{i,t+1}[j] = \mathbf{1}\{y^r_{i,t} = j'\} \min\{p_t x^v_{i,t}[j], 1\}$ where $p_t$ is sampled of r.v. with $\mathbb{E}[p_t] > 1$. $y^v_{i,t} = 1$ w.p. $x^v_{i,t}[y^r_{i,t}]$

Next, we survey papers that adopt a multi-armed bandits framework where the learner is the recommender system; the arms to pull indicates the recommendation (category) to present to the viewer; and the received reward is the feedback given by the viewer or the viewer utility. We use the notation that for any vector $a$, $a[k]$ indicates the $k$-th entry of $a$. In cases where $a$ is a one-hot vector representing the action taken by the learner, its non-zero entry represents the arm being pulled.

- In Levine et al. (2017), there are $K$ recommendation categories/arms; and $y^r_{i,t}$ is a one-hot vector of $K$ dimensions representing the recommendations given to viewer $i$ at time $t$. If $y^r_{i,t}[k] = 1$, arm $k$ is pulled. The viewer state $x^v_{i,t} \in \mathbb{N}^K_+$ has its $k$-th entry be the number of time times arm $k$ has been pulled so far. More specifically, $x^v_{i,t+1}[k] = x^v_{i,t}[k] + \mathbb{I}\{y^r_{i,t}[k] = 1\}$ and $x^v_{i,0}[k] = 0$. The expected measurement is the expected reward of pulling an arm: If $y^r_{i,t}[k] = 1$, $\mathbb{E}[y^v_{i,t}] = m_k(x^v_{i,t}[k])$ where $m_k$ is an arm-dependent monotonically decreasing function of the number of arm pulls.

- In Kleinberg & Immorlica (2018), the recommendation $y_{i,t}^r$ is defined the same as that of Levine et al. (2017). The viewer state $x_{i,t}^v \in \mathbb{N}_+^K$ has its $k$-th entry be the number of time steps elapsed since $k$ is pulled last time. More specifically, $x_{i,t+1}^v[k] = x_{i,t}^v[k] + 1$ if $y_{i,t}^r[k] \neq 1$, $x_{i,t+1}^v[k] = 1$ if $y_{i,t}^r[k] = 1$ and $x_{i,0}^v[k] = 0$. The expected measurement is the expected reward of pulling an arm: If $y_{i,t}^r[k] = 1$, $\mathbb{E}[y_{i,t}^v] = m_k(x_{i,t}^v[k])$ where $m_k$ is a concave function of the number of arm pulls. In Pike-Burke & Grunewalder (2019), $m_k$ is drawn from a Gaussian process. In Cella & Cesa-Bianchi (2020), $m_k$ is a monotonically increasing function.

- In Leqi et al. (2021), the recommendation $y_{i,t}^r$ is defined the same as that of Levine et al. (2017). The viewer state $x_{i,t}^v \in \mathbb{N}_+^K$ has its $k$-th entry be the satiation that the viewer has towards arm $k$. More specifically, $x_{i,t+1}^v[k] = \gamma_k(x_{i,t}^v[k] + y_{i,t}^r[k])$ and $x_{i,0}^v[k] = 0$ where $\gamma_k \in (0,1)$ is the satiation retention factor. The expected measurement is the expected reward of pulling an arm: If $y_{i,t}^r[k] = 1$, $\mathbb{E}[y_{i,t}^v] = b_k - \lambda_k x_{i,t}^v[k]$ where $b_k \in \mathbb{R}$ is the base reward of arm $k$ and $\lambda_k \geq 0$ is the exposure influence factor for arm $k$.

- In Ben-Porat et al. (2022), the recommendation $y_{i,t}^r$ is defined the same as that of Levine et al. (2017). The internal state $x_{i,0}^v \sim \mathbf{Q}$ is a viewer type belonging to $[B]$ ($B \in \mathbb{N}_+$) sampled from a prior distribution $\mathbf{Q}$. For $t \geq 1$, $x_{i,t}^v = x_{i,t-1}^v$ if $y_{i,t-1}^v = 1$ (i.e., viewer state remains if they have clicked on the recommendation) else $x_{i,t}^v = 0$ with probability $\mathcal{L}_{k,x_{i,t}^v}$ (indicating that the viewer may leave the platform with probability $\mathcal{L}_{k,x_{i,t}^v}$). The expected measurement is the expected click rate when the recommender pulls arm $k$, i.e., $\mathbb{E}[y_{i,t}^v] = \mathbf{P}_{k,x_{i,0}^v} \cdot \mathbb{I}\{x_{i,t}^v \neq 0\}$ if $y_{i,t}^r[k] = 1$.

- In Saig & Rosenfeld (2023), the viewer internal state $x_{i,t}^v$ is a set of tuples consisting of their previous interaction with the platform. That is, $x_{i,t}^v = \{(b_j, y_{i,j}^r, y_{i,j}^v) \mid j < t\}$ where $b_j$ is time when $j$-th event happens, $y_{i,j}^r$ is the $j$-th recommendation viewer $i$ obtained and $y_{i,j}^v$ is the viewer's reported rating. Depending on the subject of interest, $y_{i,t}^v$ is defined differently as a function of $x_{i,t}^v$. For example, $y_{i,t}^v$ can be $(b_t - b_{t-1})^{-1}$, which is viewer's "instantaneous [response] rate."

- In Laforgue et al. (2022), the recommendation $y_{i,t}^r$ is defined the same as that of Levine et al. (2017). The internal state $x_{i,t}^v$ is a $K$-dimensional vector where $x_{i,t}^v[k] = \delta_k(x_{i,t-1}^v[k], y_{i,t-1}^r)$ where $\delta_k$ is a transition function that tracks arm switches. The expected measurement when $y_{i,t}^r[k] = 1$ is $\mathbb{E}[y_{i,t}^v] = m_k(x_{i,t}^v[k])$ where $m_k$ is an arm-wise reward function. In Foussoul et al. (2023), $m_k$ is assumed to be monotonic.

- In Khosravi et al. (2023), the recommendation $y_{i,t}^r$ is defined the same as that of Levine et al. (2017). The viewer internal state is real-valued, i.e., $x_{i,t}^v \in \mathbb{R}$. If $y_{i,t}^r[k] = 1$, $x_{i,t}^v = x_{i,t-1}^v + \lambda(b_k - x_{i,t-1}^v)$ where $b_k$ is an arm-wise constant and $\lambda$ is a universal constant. If $y_{i,t}^r[k] = 1$, then $\mathbb{E}[y_{i,t}^v] = r_k \cdot x_{i,t}^v$ where $r_k \in \mathbb{R}$ is an arm-wise constant.

## C  Additional details for Content Creators in Section 3.2

The models for creator behavior can generally be broken down into two categories: models which focus on a single time step and treat creators as myopic, and models which consider non-myopic interactions across multiple time steps. In all of these models, the viewers are static.

**Myopic models.** We describe the myopic models ((Ghosh & McAfee, 2011; Ben-Porat & Tennenholtz, 2018; Ben-Porat et al., 2020; Jagadeesan et al., 2022; Hron et al., 2022; Yao et al., 2023a;b; Eilat & Rosenfeld, 2023; Immorlica et al., 2024)) where the creators (and often the recommender system) do not factor in multi-time-step interactions.

Before diving into the papers, we introduce some formalisms. Let $\mathcal{A}$ be the *action space* of the creators which captures internal decisions about what content to create. Suppose that there are $p \geq 2$ creators competing for recommendations. Each creator $j \in [p]$ has internal state $x_{j,t}^c = a_{j,t}^c$ specified by the creator's internal action $a_{j,t}^c \in \mathcal{A}$. The output $y_{j,t}^c = a_{j,t}^c$ captures the content uploaded by creator $j$, the set $y_t^c = \{y_{j,t}^c\}_{j \in [p]}$ captures the content landscape, the output $y_t^r$ captures recommendations, and the output $y_t^v$ captures viewer behaviors such as clicks.

We assume that at each time step $t$, the creator receives as *input* the recommendations $y_t^r$ and the prior content landscape $y_{t-1}^c$. The state update $f^c$ captures the creator's best-response to their utility function $U_j(a; a_{-j,t-1}^c, y_t^r)$ which depends on the action $a$ of a creator $j$, the actions $a_{-j,t-1}^c$ of other creators $j' \neq j$ in the prior content landscape $y_{t-1}^c$, and the recommendations $y_t^r$. In particular, the state update is defined by the best-response:

$$a_{j,t+1}^c = f^c(a_{j,t}^c, [y_t^r, y_{t-1}^c], w_t) := \text{argmax}_{a \in \mathcal{A}} U_j(a; a_{-j,t-1}^c, y_t^r).$$

The utility function $U_j$ (and thus the state update $f^c$) implicitly depends on the following two functions. The first function is a *creator reward function* $h(j, y_t^r, y_t^v, a)$, which captures the reward that creator $j$ receives from the measurements $y_t^r$ and $y_t^v$ along with production costs of their action $a$. The second function is the *recommender function* $g^r(x_t^r, [i, y_{t-1}^c, y_{t-1}^v], v_t)$, which maps the recommender state $x_t^r$ and inputs (i.e., the user index $i$, content landscape $y_{t-1}^c$, and viewer behaviors $y_{t-1}^v$) to recommendations $y_t^r$. In the myopic models in these papers, the recommender function $g^r(x_t^r, [i, y_{t-1}^c, y_{t-1}^v], v_t)$ only depends on the inputs $i$ and $y_{t-1}^c$ as well as the noise $v_t$. We thus introduce the simplified function $(g')^r(i; y_{t-1}^c, v_t)$ to denote recommendations. (We formalize how the utility function $U_j$ depends on $h$ and $(g')^r$ after describing each paper.)

We specify each paper in terms of how it defines the action space $\mathcal{A}$, the creator reward function $h$, and the recommender function $(g')^r$:

- Ghosh & McAfee (2011) take $\mathcal{A} = [0, 1]$ (capturing quality). The recommender function $(g')^r$ awards prizes to creators based on their quality rank. The reward $h$ captures prize for creator $j$ according to the recommendation ranks given by $y^r$ minus a 1-time cost of production $c(a)$. The model additionally incorporates participation decisions.

- Ben-Porat & Tennenholtz (2018) take $\mathcal{A}$ to be a finite set. The recommender function is a randomized function $(g')^r$ mapping each viewer to a content in $y^c$ or no content (denoted by $\emptyset$). The reward $h$ captures the number of recommendations in $y^r$ assigned to creator $j$.

- Ben-Porat et al. (2020) take $\mathcal{A}$ to be a finite set capturing content topics. When a given creator $j$ writes on a topic $a \in \mathcal{A}$, their article is a predetermined quality $q_{a,j}$. Each viewer seeks a particular topic $a \in \mathcal{A}$ of content. The recommender function $(g')^r$ assigns a viewer seeking a topic $a$ the content with highest quality of that topic: that is, $\text{argmax}_{j \in [p]} q_{a,j} \cdot I[a = a_j^c]$. The reward $h$ captures the number of recommendations in $y^r$ assigned to creator weighted by topic-specific weights.

- Jagadeesan et al. (2022) take $\mathcal{A} = \mathbb{R}_{\geq 0}^D$ to be $D$-dimensional content embeddings in the nonnegative orthant. Each viewer $i$ has a fixed preference vector $u_i \in \mathbb{R}_{\geq 0}^D$. The recommender function $(g')^r$ assigns the viewer $i$ the content created by creator $j$ that maximizes the inner product $\langle u_i, a_j^c \rangle$: that is, $\text{argmax}_{j \in [p]} \langle u_i, a_j^c \rangle$. The reward $h$ is the number of recommendations won minus the 1-time cost of production specified by $c(a) = ||a||^\beta$.

- Hron et al. (2022) take $\mathcal{A}$ to be $D$-dimensional content embeddings in the unit sphere $\{x/||x||_2 \mid x \in \mathbb{R}^D\}$. Each viewer $i$ has a fixed preference vector $u_i \in \mathbb{R}^D$. The recommender function $(g')^r$ assigns viewer $i$ the content created by creator $j$ with probability proportional to $e^{\eta \langle u_i, a_j^c \rangle}$. The reward $h$ is the number of recommendations won.

- Yao et al. (2023a) take $\mathcal{A} \subseteq \mathbb{R}^D$ to be an abstract set. There is a set of viewers $X \subseteq \mathbb{R}^D$. The recommender system computes scores for each content and viewer pair and assigns the top $K$ recommendations to be the top $K$ scores. The reward $h$ is the sum of the scores of the recommendations won.

- Yao et al. (2023b) take $\mathcal{A} \subseteq \mathbb{R}^D$ to be an abstract set. There is a set of viewers $X \subseteq \mathbb{R}^D$. The recommender system computes scores for each content and viewer pair and determines recommendations $(g')^r$ based on an arbitrary function of these scores and the viewer characteristics. The reward $h$ captures the reward received by the creator (specified by a general reward function of the scores and the viewer) minus the production cost $c(a)$.

- Immorlica et al. (2024) take $\mathcal{A} = \mathbb{R}_{\geq 0}^2$ to be 2-dimensional content embeddings capturing a clickbait dimension and quality dimension. Each viewer $i$ has a fixed type $s_i > 0$ representing their tolerance for

clickbait and only engages with content if they derive nonnegative utility (viewer utility increases with quality and decreases with clickbait). The recommender function $g^r$ optimizes an engagement metric (which is misaligned with viewer welfare) and assigns viewer $i$ the content that maximizes viewer $i$'s engagement. The reward $h$ is the number of recommendations won (that viewers actually engage with) minus a 1-time cost of production.

We specify the form of the creator utility function $U_j$. The creator utility function $U_j$ anticipates the impact of the creator's actions on recommendations in the next time step, which make it slightly messy to formalize in the dynamical system framework, but which we can nonetheless formalize as follows. At time step $t$, creators $j$ first assesses how their actions $a$ and the actions of other creators $a^c_{-j,t-1}$ would affect the content landscape

$$\tilde{y}^c_t(a) := [a^c_{1,t-1}, a^c_{2,t-1}, \ldots, a^c_{j-1,t-1}, a, a^c_{j+1,t-1} \ldots, a^c_{p,t-1}].$$

Then, the creator assesses the impact on the downstream recommendations

$$\tilde{y}^r_{t+1}(a) := [(g')^r(1; \tilde{y}^c_t(a), v_{t+1}), \ldots, (g')^r(n; \tilde{y}^c_t(a), v_{t+1})]$$

and observed viewer behaviors

$$\tilde{y}^v_{i,t+1}(a) = g^v(x^v_i, \tilde{y}^r_{i,t+1}(a), v_{t+1}),$$

and computes their utility:

$$U_j(a; a^c_{-j,t-1}, y^r_t) := h(j, \tilde{y}^r_{t+1}(a), \tilde{y}^v_{t+1}(a), a).$$

Recall that the papers described above study the equilibrium in the game between creators, which correspond to *fixed points* of these dynamics. More specifically, the fixed points of the dynamics described above correspond to the *pure Nash equilibria*. Several papers also consider *mixed Nash equilibria*, but expressing these equilibria as fixed points requires introducing randomness into the dynamics and carefully accounting for who observes the realizations of the randomness.

We note that some myopic models do not fit into structure that we described above. For example, Eilat & Rosenfeld (2023) models the recommender system as choosing (or learning) user embeddings. The user embeddings affect recommendations through a fixed ranking function and a fixed bipartite relevance graph. Each creator chooses the content that maximizes the average (linear) relevance score for embeddings of users connected to them in the relevance graph.

**Non-myopic models.** We next turn to the non-myopic models (e.g. Ghosh & Hummel, 2013; Hu et al., 2023; Buening et al., 2023), where creators and the recommender system factor in multi-time-step interactions. These models of content creator behavior are generally more complicated to formulate. The recommender function $g^r$ is modelled as a multi-bandit algorithm. Creators account for their (potentially discounted) cumulative reward when selecting actions.

- Ghosh & Hummel (2013) consider a setup where each creator $j$ enters the platform at a potentially different time step. The action set is $\mathcal{A} = [0,1]$ and captures content quality. The recommender system is a stochastic multi-armed bandit algorithm. The reward $h$ is the number of recommendations to be won at the current round and in the future minus the cost of production $c(a)$. The model additionally incorporates participation decisions.

- Hu et al. (2023) consider a setup where all creators arrive at every time step and can create new content at each time step. The action set is $\mathcal{A} \in (\mathbb{R}^D_{\geq 0})^T$. The recommender system is an adversarial multi-armed bandit algorithm. The reward $h$ is the cumulative discounted sum of the number of recommendations to be won minus the cost of production at each time step. Production costs at each time step $t$ are specified by $c(a) = ||a||^\beta$. This model makes the simplifying assumption that each creator chooses the content that they will produce at every time step at the beginning of the game.

- Buening et al. (2023) consider a setup where creators all arrive at every time and each choose the feedback rate $a \in \mathcal{A} = [0,1]$ of their content at the beginning of the game. The recommender system is a multi-armed bandit algorithm that accounts for probabilistic feedback. The reward $h$ is the total number of recommendations won.

