# OpenReview forum: "Accounting for AI and Users Shaping One Another: The Role of Mathematical Models"
_TMLR — Accepted by TMLR_

### Review · Reviewer_q8D4 · 2025-01-26

**Summary Of Contributions:**

In computer science and machine learning, it is very common to model and study systems in a way that doesn't take the complexities of their interactions with humans into account. In the field of human computer interaction, it is very common to try taking these interactions into account, but most work fails to address the possibility that, through interaction, the system and user can change. For example, after interacting with a recommendation algorithm, my content preferences may change overtime. This is a survey paper (which is lowkey a little bit of a position paper) that is dedicated to the use of human/computer models that allow for change over time in HCI. Most of it is dedicated to surveying and discussing these types of models, including how/why they can be used as well as a case study into the broad challenge of studying content recommendation.

**Audience:**

Yes

**Claims And Evidence:**

Yes

**Requested Changes:**

Above, I asked the authors to consider a few things. Please consider them. However, overall, I think it is obvious that this paper is really thorough and well-written. I think it does what it has set out to do very well. If I knew the literature behind this work better, I might have some specific advice for how to improve discussion of specific work. But I do not. To me, this paper seems ready as is with potentially minimal changes.

Overall, I think this paper has the potential to be valuable. And most of all, I just think it has been well executed. I will recommend acceptance unless other reviewers have major concerns.

**Strengths And Weaknesses:**

S1: I like the premise of the paper. We often treat AI systems as standalone things. But in the real world, they interact with humans (and other AI systems) and the dynamics behind those interactions might change over time.

S2: The writing and its flow are great. It's almost a pleasure to read.

S3: I think the engagement with prior work seems pretty thorough and does not seem perfunctory. I'm not an expert unrelated literature so I can't comment on whether or not things are missing. But it looks like the authors did their homework.

W1: The paper is lacking an illustrative & pretty figure 1 to convert the main couple ideas of the paper. Consider adding one if appropriate.

W2: the central example of content recommendation seems to highlight another challenge with modeling humans, and a shortcoming of past works' modeling of humans. Humans have different orders of preferences. For example, my lizard brain might want me to keep scrolling through cat videos though another part of my brain is telling me to watch educational content. It seems a little bit odd to address only one of these problems at a time. For a problem like this, we need to consider the fact that human preferences can change, but we also need to be wary of the complex, multifaceted, and contextual nature of human preferences themselves. Overall, I don't think this is really a problem with the contribution of the paper, but it reflects a slight uneasiness I have with the way the paper discusses things and motivates using examples.

W3: this isn't really much of a weakness. I think this paper is a really good survey paper and will be valuable. But I usually think that survey papers are a bit better if they do something on top of the survey, such as some quantitative analysis about what's going on in the field, some experiments, etc. Again, this isn't a real weakness for me. But if the authors can think of any useful things to add, that might be interesting.

---

> ### Author Response · Authors · 2025-04-16
>
> Thank you for your kind review and thoughtful suggestions! We have made several changes in the revision. In particular, based on your suggestions, we included additional summary figures to illustrate our main ideas (section 2.2) and a table to summarize our analysis (section 3).

---

### Review · Reviewer_u8TQ · 2025-03-06

**Summary Of Contributions:**

This paper introduces the concept of formal interaction models, mathematically specified as coupled dynamical systems, to explicitly capture how AI systems and users reciprocally shape each other's states and behavior. The authors articulate four key use-cases for these models: (1) specifying interactions for implementation, (2) monitoring real-world interactions empirically, (3) anticipating societal impacts through counterfactual analyses, and (4) controlling societal impacts via algorithmic or policy interventions. They provide a thorough and structured overview of the design considerations for these models, inluding style, granularity, mathematical complexity, and measurability. A case study of content recommendation systems illustrates and critically assesses existing literature and underscores both the utility and limitations of formal interaction models.

**Audience:**

Yes

**Claims And Evidence:**

No

**Requested Changes:**

### Empirical Validation (Critical):

Include empirical validation or illustrative simulation results, using real-world datasets or established simulation environments (e.g., RecSim, recogym). Demonstrating how formal interaction models can empirically capture or mitigate unintended societal impacts would substantiate theoretical claims.

### Enhanced Comparison to Existing Approaches (Important):

Clearly position the contribution relative to established frameworks such as model-free and model-based RL, causal inference, or economic models. Consider providing a brief comparative table or figure to clarify the added value and differentiation of formal interaction models from these established frameworks.

### Detailed Guidelines on Practical Implementation (Moderate):

Provide explicit examples or scenarios illustrating how to practically navigate the trade-offs along the axes of granularity, measurability, and complexity.

Consider adding concrete examples or short case illustrations in a dedicated appendix to clarify model design decisions clearly.

**Strengths And Weaknesses:**

Strengths:

- Structured and comprehensive approach: The paper delineates multiple practical use-cases for formal interaction models (implementation, empirical monitoring, societal impact anticipation, and control). The outlined framework is precise and helps bridge theory and practical application.

- Insightful case study analysis: The authors analyze the existing literature on recommender systems, highlighting gaps and future research opportunities. By applying the framework, they illustrate the potential utility of formal interaction models.

Weaknesses:

- Limited empirical validation: The theoretical nature of the paper lacks empirical demonstrations to showcase practical effectiveness.

- Insufficient integration with existing frameworks: The paper doesn't clearly position itself relative to established simulation and modeling methods (e.g., reinforcement learning environments).

- Abstract practical guidelines: The advice on navigating trade-offs regarding measurability and complexity lacks concrete examples or clear guidelines.

---

> ### Author Response · Authors · 2025-04-16
>
> Thank you for your thoughtful feedback.
>
> We have made several changes in the revision. In particular, based on your suggestions and requests, we added clarifications to enhance the comparison with related frameworks like RL and causal inference (introduction and section 2.2); we revised our running example (section 2.1) and use it to concretely discuss the design axes (section 2.3); we included a new summary figure on design axes and use cases (section 2.2).
>
> We clarify that this is a survey paper rather than a methods paper. Several of the papers included in our analysis present empirical demonstrations of individual formal interaction models, and our goal is to summarize and synthesize these works. As such, we consider an empirical investigation out of the scope of our work. Given our clarification, are you willing to reconsider this requested change?

---

### Review · Reviewer_2QUS · 2025-03-23

**Summary Of Contributions:**

This survey paper pushes the AI community for the need to address various distribution shifts (user preferences, content distributions, etc) by formal mathematical modeling.
P.S: This is a short review as I am in an emergency situation myself.

**Audience:**

Yes

**Broader Impact Concerns:**

-

**Claims And Evidence:**

Yes

**Requested Changes:**

-

**Strengths And Weaknesses:**

This work discusses a great number of use cases for this problem. I think incorporating the current methodologies (off-policy learning, contrastive methods, etc) for solving this problem, and discussing their limitations, and probable future steps, will further improve the contributions in this work.

---

### Review · Reviewer_dYcT · 2025-03-25

**Summary Of Contributions:**

This paper argues for the development of formal interaction models -- a formalization of interaction between human users and an AI system. These models are based on the framework of coupled dynamical systems, which the authors suggest to serve as a unifying system to represent all sorts of interaction models. The authors focus on recommender systems to illustrate the functioning and benefits of formal interaction models, building examples that include a recommendation system, a viewer, and a creator, which reciprocally influence each other. Users' time spent is understood as the input to the recommendation system, while suggested content is understood as it's output, each serving to measure and update states in user and system. The authors posit that formal interaction models can be used for four main points: specifying user interactions, monitoring interactions, anticipating societal impacts, and controlling societal impacts. The authors add a review of literature investigating viewer preferences and creator content and cast both concepts into formal interaction models.

**Audience:**

Yes

**Broader Impact Concerns:**

There was only one point in the manuscript where I briefly noted broader impact concerns. In Section 2.3, measurability is discussed as a design axes of formal interaction models. The last sentence in this paragraph already hints at the possibility that some concepts are unobservable. Dysfunctional AI systems are often based on the assumption that aspects such as psychological well-being or employability are easy to quantify and predict. I would like a brief description of how formal interaction models would deal with these unobservable constructs and what the (maybe ethical) implications for their designs are. To this end, I recommend: Jacobs, Abigail Z., and Hanna Wallach. "Measurement and fairness." Proceedings of the 2021 ACM conference on fairness, accountability, and transparency. 2021.

**Claims And Evidence:**

Yes

**Requested Changes:**

Note that these are suggestions:
- Introduction of a running example to illustrate the claimed use cases of formal interaction models
- Inclusion of a methods section or similar to outline survey approach
- Discussion of how complex user preferences and adverse system effects are represented in formal interaction models, see for example [1,2,3]
- Discussion to address relation between formal interaction models and societal impact
- Inclusion of a conclusion
- More details or examples of different models' styles and granularities
- Clarification on how the state update and measurement equation in Example 1 were defined
- Grammar and typography check, I found several word repetitions and misspellings
- Workover of the second-to-last paragraph on page 9, here the text is hard to parse due to the references

[1] Shulner-Tal, Avital, Tsvi Kuflik, and Doron Kliger. "Fairness, explainability and in-between: understanding the impact of different explanation methods on non-expert users’ perceptions of fairness toward an algorithmic system." Ethics and Information Technology 24.1 (2022): 2.
[2] McInerney, James, et al. "Explore, exploit, and explain: personalizing explainable recommendations with bandits." Proceedings of the 12th ACM conference on recommender systems. 2018.
[3] Schmude, Timothée, et al. "Information that matters: Exploring information needs of people affected by algorithmic decisions." International Journal of Human-Computer Studies 193 (2025): 103380.

**Strengths And Weaknesses:**

## Strengths
The paper is well-written, clearly structured, and makes a clear claim for its contribution. The linguistic style is accessible and easy to follow, the mathematical formulations are to the point and can be grasped quickly. The paper introduced to me the idea of formal interaction models, which I found interesting. The paper uses several examples to guide understanding and identifies several implications and avenues for future work. All in all a thorough and pleasant paper.

## Weaknesses
While the paper's claims regarding how formal interaction models can be used and what they can achieve are not implausible, I was wondering multiple times how these aims should be fulfilled precisely. As an example, the introduction states that formal interaction models can, in extension, be useful to build reliable AI systems with positive social impact. Examples of applications are "building, evaluating, deploying, and auditing AI systems". However, after reading the paper, I do not necessarily have a clearer idea of how formal interaction models can be used to build reliable AI systems nor can I picture them in, e.g., AI auditing. The paper demonstrates how formal interaction models can be used to formalize user boredom with respect to a recommender system, but how does this translate into societal impact? Similar questions arose regarding the "use cases" of formal interaction models (specifying, monitoring, anticipating, controlling). While the cited literature reflects all these aspects, it did not become clear to me how the formalization of interaction supports these use cases specifically. Here, I feel that a more thorough running example would improve the tangibility of concepts and claims.

Related to this, I did not get a clear picture of the level of abstraction in which formal interaction models are suggested to be applied. (Although here as well I was wondering how the exact terms came to be defined, such as the state update 0.5 * x + y.) I can imagine a recommender system employing the user boredom model as an input to its content ranking. In contrast, I find it difficult to picture how and which formal interaction model can represent polarization and addiction (Section 3.2). Addiction, for example, seems to include many variables that would need to be considered in this model, such as psychological conditions (addictive tendencies, social environment), interaction features (infinite scrolling), and type of content. Do the authors suggest that these aspects are captured with noise variable w (this variable seems to be underspecified)? Or are formal interaction models not supposed to represent concepts like addiction in detail? In short, the paper does not fully provide enough argumentative ground or examples to grasp how formal interaction models impact society-level issues. Here, I invite the authors' clarification.

Parts of these questions might also be answered through an extension of Section 2.3, which describes design choices for formal interaction models. On the one hand, I wondered how these design axes were selected, and on the other, I would appreciate more concrete examples of how these models can have different styles and granularities. Another suggestion to address these questions would be the extension of the discussion, which currently is quite short.

Lastly, I am missing a Section that describes how papers for the literature survey were selected and analyzed. While I understand that the paper does not claim that it conducted a systematic literature review, some information on how the concepts presented were synthesized would be welcome.

---

> ### Author Response · Authors · 2025-04-16
>
> Thank you for your thorough feedback.
>
> Our revision includes many changes. In particular, based on your feedback, we revised the running example (section 2.1) and used it to concretely discuss the design axes (section 2.3). We also added further clarification on individual vs. population level effects and societal impacts (section 2.1). We included your suggested reference on measurement with a discussion of unobservability (section 2.3). We added a description of how we collected papers for our analysis (section 3). Do these changes address your concerns, and are there any remaining changes you would like to see?

---

### Author Response · Authors · 2025-04-04
**extend discussion period**

We thank the reviewers for their thoughtful comments and the action editor for facilitating the process! We wanted to repeat a request that we made last week: Would it be possible to extend the discussion period by at least one week, until April 13th, and preferably by two weeks, until April 20th? Due to prior commitments (travel, etc), my collaborators and I aren't able to fully engage in discussions until next week.

---

### Author Response · Authors · 2025-04-16
**new revision uploaded**

Thanks to the reviewers for their feedback, and to the AE/EIC for facilitating the process! We have uploaded a new revision (changes listed below) based on reviewer comments and questions. We hope to actively engage in discussion on any remaining concerns over the next several days.

Changes since initial submission:
- Clarifying connection/comparison with RL and causal modelling (introduction, section 2.2)
- Clarifying individual vs. societal level impacts (section 2.1)
- Revised simple example to better illustrate connection between formal model and societal impacts (section 2.1)
- New summary figure on design axes and use cases (section 2.2)
- More detail on game theoretic model style (section 2.3)
- Further comment on measurability and unobservability (section 2.3)
- Illustration of design axes with simple example (section 2.3)
- New summary table on literature review (section 3)
- More detail on inclusion criteria for papers in our analysis (section 3)
- Minor changes to wording, typos, etc (throughout)

---

### Decision · Action_Editor_sgCB · 2025-05-25

**Recommendation:** Accept as is

**Comment:**

The paper received four reviews containing several suggestions for improvements (mainly regarding clarifications, e.g., adding/expanding on running examples, and expanding on the relation to other modeling methods, e.g., reinforcement learning environments). The authors have updated their paper in line with these suggestions and all four reviewers recommended the acceptance of the paper. Thus, as the claims of the paper are substantiated and the paper is relevant to the TMLR audience, I am recommending the acceptance of the paper.

**Audience:**

Yes, the paper can be valuable to those developing AI systems and those aiming to understand their impact better.

**Claims And Evidence:**

The submitted paper is mainly a survey paper and provides a good overview of the literature regarding the usage of formal interaction for better developing and understanding AI systems. The claims regarding the importance of formal interaction models are well-supported by examples and connected to the existing literature.